Microsatellite loci development and population genetics in Neotropical fish Curimata mivartii (Characiformes: Curimatidae)

Landínez-García Ricardo M.
Marquez Edna J. ejmarque@unal.edu.co
Laboratorio de Biología Molecular y Celular, Facultad de Ciencias, Escuela de Biociencias, Universidad Nacional de Colombia—Sede Medellín , Medellín , Colombia
Amorim Antonio
Electronic publication date: 2018 Nov 13
Publication date: 2018
Volume: 6
Electronic Location ID: e5959
Received 2018 Mar 26; Accepted 2018 Oct 18
Copyright: © 2018 Landínez-García and Marquez
Copyright year: 2018
Copyright holder: Landínez-García and Marquez
License: This is an open access article distributed under the terms of the Creative Commons Attribution License, which permits unrestricted use, distribution, reproduction and adaptation in any medium and for any purpose provided that it is properly attributed. For attribution, the original author(s), title, publication source (PeerJ) and either DOI or URL of the article must be cited.
License URL: https://creativecommons.org/licenses/by/4.0/

Keywords: Molecular marker, Freshwater fish, Gene flow, Outlier loci, Genetic diversity

Funding: Universidad Nacional de Colombia and Integral S.A., on 19th September 2013 Universidad Nacional de Colombia, Sede Medellín and Empresas Públicas de Medellín CT-2013-002443 Variación genotípica y fenotípica de poblaciones de especies reófilas presentes en el área de influencia del proyecto hidroeléctrico Ituango This work was supported by the Universidad Nacional de Colombia and Integral S.A., on 19th September 2013 and Universidad Nacional de Colombia, Sede Medellín and Empresas Públicas de Medellín, Grant CT-2013-002443 “Variación genotípica y fenotípica de poblaciones de especies reófilas presentes en el área de influencia del proyecto hidroeléctrico Ituango.” The funders had no role in study design, data collection and analysis, decision to publish, or preparation of the manuscript.

==============================
The Curimatidae family plays an ecological role in the recycling and distribution of nutrients and constitutes a major food source for several commercially important fishes. Curimata mivartii, a member of this family, is considered a short-distance migratory species (≤100 km), categorized by the International Union for Conservation of Nature as a near threatened species, based on its declining population densities and habitat disturbance and fragmentation. Since population genetics and species-specific molecular tools remain unknown for all members of the Curimatidae family, this study developed a set of microsatellite loci and studied the population genetics of C. mivartii in the lower section of the Colombian Magdalena-Cauca basin. The results showed high levels of genetic diversity and evidence of gene flow even between locations separated over 350 km. This information provides a baseline for designing conservation and management programs for C.mivartii and constitutes the first study of population genetics in Curimatidae.

Introduction

The Curimatidae (Pisces: Characiformes) family encompasses eight genera and approximately 105 species that present a wide distribution in the freshwater environments of the cis- and trans-Andean basins of South America (Melo et al., 2016a). This is the fourth most diverse family of the Characiformes order, and its members have been increasing in number over the last 20 years due to the discovery and introduction of new species (Melo et al., 2016a). Although their commercial importance is limited to the subsistence fisheries and ornamental species trade, these detritivorous, benthopelagic and migratory species contribute to the recycling and redistribution of nutrients (Alvarenga et al., 2006). Additionally, they represent higher percentages of biomass, providing food to birds and a large variety of fishes, particularly economically important catfish species, that support the nutritional safety of the riverside communities (Lasso et al., 2010).

Curimata mivartii Steindachner 1878, “vizcaína” or “cachaca” is a Colombian endemic species that represents the dominant biomass of the floodplain lakes in the middle basin of the Magdalena, San Jorge and Sinú rivers (Fig. 1). It is the largest Colombian trans-Andean species of the genus (approximately 35 cm) and has a short-distance migration range (approximately 10.1 km; average speed 0.3 km/day) (Lasso et al., 2010; López-Casas et al., 2016) through the main channel of the river. It forms great shoals and uses floodplain lakes as habitats for nourishment, refuge and larval development (Lasso et al., 2010). Like other species (Jiménez-Segura, Palacio & Leite, 2010; Zapata & Usma, 2013), its migrations seem to be related to changes in the water level of the rivers, which are bimodal in the Magdalena river basin (Jiménez-Segura, Palacio & Leite, 2010).

Figure 1 Photography of Curimata mivartii, a freshwater fish endemic to Colombia.

During migrations, C. mivartii is mainly exploited by subsistence fisheries, but it has become a target of commercial fishing due to decreasing capture of traditional species (Mojica et al., 2012). The estimated decline of 30% of its population density and the noticeable fragmentation of its habitats caused by the growing anthropic intervention have led to the inclusion of C. mivartii in the “Near Threatened” list, according to the international union for conservation of nature. Additionally, biological information about the species is scarce and the population genetics is unknown for all members of the Curimatidae family, which limits the design of appropriate conservation and management programs.

In contrast, population genetics studies have been performed on members of the phylogenetically related family, Prochilodontidae (Melo et al., 2016b; Vari, 1989), with which Curimatidae shares habitats and migratory, benthopelagic and detritivorous features. These studies reveal that Prochilodontidae exhibits populations formed by both nongeographical (Hatanaka, Henrique-Silva & Galetti, 2006; López-Macías et al., 2009; Melo et al., 2013; Orozco Berdugo & Narváez Barandica, 2014) and geographically structured stocks (Landínez-García & Márquez, 2016).

Given the genetic structure of Prochilodontidae populations and the short-distance migration described for C. mivartii, we hypothesized that their populations are structured according to an isolation by distance model (Wright, 1943). To test this hypothesis, this study developed a set of primers for microsatellite loci amplification and evaluated the genetic diversity and structure of C. mivartii samples from different locations along the main stream and some floodplain lakes of the Magdalena-Cauca basin, encompassing differences in distance extending to over 350 km between the farthest locations.

Materials and Methods

This study analyzed a total of 209 muscle tissues of C. mivartii from the main stream of the rivers and floodplain lakes in the lower section of the Colombian Magdalena-Cauca basin. The study area includes floodplains of the Andean Magdalena-Cauca basin that present riverbeds with an upper width of 500 m, low velocities, rock shards and fine sediments. This area comprises a group of floodplain lakes, which are the principal habitats of C. mivartii and differ in size, depth, levels of connection and anthropic intervention (farming and cattle breeding). Additionally, during the last decade (between 1997/1998 and 2009/2010) the Magdalena-Cauca basin has been exposed to the remote more extreme effects of atypical climate fluctuations such as the El Niño Southern Oscillation (ENSO) events (IDEAM—Instituto de Hidrología Meteorología y Estudios Ambientales, 2014), with eight to nine successive months (NOAA’s Climate Prediction Center, 2015) of extreme floods (La Niña) and droughts (El Niño) in trans-Andean fluvial systems (Bookhagen & Strecker, 2010).

All samples, preserved in 70% ethanol, were provided by Integral S.A., through two scientific cooperation agreements (September 19th, 2013; Grant CT-2013-002443). Sampling collection was performed from 2011 to 2013 by Integral S.A., framed under an environmental permit from Ministerio de Ambiente, Vivienda y Desarrollo Territorial de Colombia # 0155 on January 30, 2009 for Ituango hydropower construction. The first group of samples was collected from the Cauca river (Fig. 2A), three floodplain lakes (Grande, Las Culebras, Panela—Bolívar Department), a site on the Caribona river (La Raya, San Jacinto del Cauca—Bolívar Department) and another site on the Man river (Caucasia—Antioquia Department). The second group of samples was collected from the Magdalena river (Fig. 2B), a floodplain lake (Chucurí, Puerto Parra—Santander Department) and a site on the main channel of the river (Puerto Berrío, Antioquia Department).

Figure 2 Sampling sites (star) of Curimata mivartii in some sectors of the Colombian rivers Cauca (A) and Magdalena (B).

To develop species-specific microsatellite loci, the sequence reads from the genomic library of one specimen of C. mivartii, previously pyrosequenced by 454 FLX technology (Landínez-García, Alzate & Márquez, 2016), were analyzed using PRINSEQ lite software to eliminate sequences of less than 100 pb in length, duplicated reads and low-quality regions. Then, PAL_FINDER v.0.02.03 (Castoe et al., 2010) was used to extract the reads that contained perfect tri-, tetra- and pentanucleotide microsatellite motifs, Primer3 v.2.0 (Rozen & Skaletsky, 2000) for the primer designs in the flanking sequences of the microsatellite loci and ePCR (Rotmistrovsky, Jang & Schuler, 2004) for assessing the correct alignment of the primers. The level of polymorphism was evaluated in 40 microsatellite loci and further genetic analyses were performed with a set of loci that satisfied the characteristics required to validate new microsatellite primers (Neff, Garner & Pitcher, 2011; Fernandez-Silva et al., 2013; Schoebel et al., 2013).

For genotyping, the PCR mixtures were performed in volumes of 10 μl containing the following final concentrations: 0.6 pmol/μl of each forward primer tagged on the 5′ end with one of the adapters described by Blacket et al. (2012), 12 pmol/μl of each reverse primer, 10 pmol/μl of each fluorescently labeled adapter, 1.1× Master Mix, 2.5% v/v GC Enhancer Platinum Multiplex PCR Master Mix (Applied Biosystems, Foster City, CA, USA) and three to five μg/μl of template DNA isolated with GeneJET DNA purification kit (Thermo Scientific, Waltham, MA, USA), following the manufacturer’s instructions. The amplification reactions were performed on a thermocycler T100 (BioRad, Hercules, CA, USA) with an initial denaturation step of 95 °C for 3 min, followed by 42 cycles consisting of a denaturation step of 90 °C for 22 s, an annealing step of 56 °C for 20 s and without an extension step or final elongation. The amplicons were separated by electrophoresis on an ABI 3730 XL automated sequencer (Applied Biosystems, Foster City, CA, USA) using LIZ500 (Applied Biosystems, Foster City, CA, USA) as the internal molecular size. Then, GeneMapper v.4.0 (Applied Biosystems, Foster City, CA, USA) was used to denote and score the allelic fragments according to their molecular size and Micro-Checker v.2.2.3 (Van Oosterhout et al., 2004) was run to detect potential genotyping errors.

The average number of alleles per locus and the polymorphism information content (PIC) for each marker were calculated respectively using GenAlEx v.6.503 (Peakall & Smouse, 2006) and Cervus v.3.0.7 (Marshall et al., 1998). The average number of alleles per locus, observed (HO) and expected (HE) average heterozygosities and fixation index were calculated to estimate the genetic diversity of C. mivartii. Tests for departures from Hardy–Weinberg and linkage equilibria as well as the HO and HE were estimated using Arlequin v.3.5.2.2 (Excoffier, Laval & Schneider, 2005). The sequential Bonferroni correction was applied to adjust the statistical significance in multiple comparisons (Rice, 1989).

To explore nonneutral evolutive forces acting on the microsatellite loci, scanning analyses were performed using BayeScan v.2.1 (Foll & Gaggiotti, 2008) software to detect diversifying, positive or balancing selection. Parameters included 10:1 prior odds for the neutral model, 20 pilot runs consisting of 5,000 iterations each, followed by 200,000 iterations and a burn-in of 50,000 iterations. The loci were ranked according to their estimated posterior probability or posterior odds (equivalent to the Bayes factor), a statistical criterion to test the model. Positive alpha values were then used to distinguish microsatellites under diversifying selection, while negative alpha values were used to detect balancing selection (Foll & Gaggiotti, 2008).

To explore whether outlier loci found in this study represented false positives, the ocurrence of recent genetic bottlenecks of populations was evaluated by calculating the levels of heterozygosity using the Wilcoxon sign-rank test (Luikart & Cornuet, 1998) included in Bottleneck v.1.2.02 software (Piry, Luikart & Cornuet, 1999). Addittionally, the M ratio (the mean ratio of the number of alleles compared with the range in allele size) was calculated using Arlequin v.3.5.2.2 (Excoffier, Laval & Schneider, 2005). The M ratio indicates that the population has experienced recent and severe reduction in population size when its values are smaller than 0.68 (Garza & Williamson, 2001).

After exploring the genetic structure of the samples, the hypothesis of isolation by distance was tested using a Mantel test (Mantel, 1967) implemented in GenAlex v.6.503 (Peakall & Smouse, 2006). The genetic structure among geographical samples was calculated using the standardized statistics F′ST (Meirmans, 2006) and Jost’s Dest (Meirmans & Hedrick, 2011), analysis of molecular variance (AMOVA) (Meirmans, 2006), with 10,000 permutations and bootstraps included in GenAlex v.6.503 (Peakall & Smouse, 2006). Patterns of genetic structure were further explored, using the diploid genotypes of 20 loci (40 variables) in 209 individuals, which were submitted to discriminant analysis of principal components using the R package Adegenet (Jombart, 2008).

Likewise, Bayesian analysis of population partitioning implemented in STRUCTURE v.2.3.4 (Pritchard, Stephens & Donnelly, 2000) was used to examine other sample groupings. Parameters included 350,000 Monte Carlo Markov Chain steps and 20,000 iterations such as burn-in, admixture model, correlated frequencies and the LOCPRIOR option for detecting relatively weak population structures (Hubisz et al., 2009). Each analysis was repeated 20 times for each simulated K-value, which ranged from 1 to 10 groups. For the best estimation of genetic stocks (K), the ΔK ad hoc statistic (Evanno, Regnaut & Goudet, 2005) was calculated using STRUCTURE HARVESTER (Earl & VonHoldt, 2012). Then, CLUMPP v.1.1.2b (full-search algorithm, function G′normalized, parameters at their default values) (Jakobsson & Rosenberg, 2007) and DISTRUCT v.1.1 (Rosenberg, 2004) were used, respectively, to summarize the results of independent STRUCTURE runs and plot the Q-matrix obtained in a histogram, displaying the ancestry of each individual in each population.

Results

A set of 27 out of the 40 loci evaluated was selected as they showed clearly defined peaks, the absence of stutter bands, nonspecific bands, high polymorphism, reproducibility and correct motif sizes, among other parameters proposed (Neff, Garner & Pitcher, 2011; Fernandez-Silva et al., 2013; Schoebel et al., 2013). These microsatellite loci include penta- (4), tetra- (8) and trinucleotide (15) motifs, which exhibited allele lengths ranging from 90 to 350 bp and PIC values ranging from 0.549 to 0.946 (average: 0.791) (Table 1). The number of alleles per locus ranged from 6 to 23 (average: 10.493) and the average levels of heterozygosity across loci and samples were HO = 0.757 and HE = 0.801 (Table 2). These genetic diversity parameters showed the highest values in the floodplain lake Las Culebras (10.900 alleles per locus; HO = 0.757; HE = 0.793) and the lowest values in La Raya (9.650 alleles per locus; HO = 0.734; HE = 0.803) (Table 2).

Table 1 Primer sequences and characteristics of 27 microsatellite loci selected for Curimata mivartii.

Locus name	Primer sequence for forward (F)and reverse (R) (5′−3′)	Repeat motif	Size range (bp)	PIC	
Cmi01	F: TTGGGTTAATGTATAGGTACAGATTGG	(ATCT)n	124–228	0.928	
R: TTTGAACATGCAACTTTGAGC		
Cmi02	F: ATAGCACCCACAGCCACTCC	(ATCT)n	175–279	0.941	
R: AAGGTGGCCTGTTCATCAGC		
Cmi03	F: CCTCATTACATTGTATGCAACAGC	(CAT)n	290–305	0.679	
R: ATTTGGAGACCTCGTGGTGG		
Cmi05	F: CATCCTGCACGTTTACACTCC	(ATCT)n	242–318	0.895	
R: CATTGTTTAAAACCTGGGAGC		
Cmi07	F: CTGTTTGAGTTCTGAGTTTTGGC	(ATCT)n	264–328	0.856	
R: CGTTTGGCATAATGGTCACG		
Cmi08	F: GAAGGACATCTGGAGGAGAAGG	(AATAG)n	226–306	0.722	
R: AAGAAAAGGCACGCTTGTGG		
Cmi09	F: ACAGCGCTGACTGTCCACC	(AGAGC)n	90–130	0.664	
R: TTGTGGATCTGCTCGTGTCC		
Cmi11	F: TGCTAAGAAGCCCTGAACTGG	(ATT)n	213–264	0.872	
R: GAATCGCAGTGAACCACACG		
Cmi12	F: GAAAGCTGGATGGATTTGGC	(ATT)n	205–244	0.766	
R: ATTAAGCGGGTTTGACGACG		
Cmi16	F: CTCTCTGTATTATGCCCGAGTCC	(ATT)n	179–215	0.812	
R: GCACCAACACCTTGCTACCC		
Cmi17	F: GCCAACTAAGCTAACCAACAAGC	(AAAT)n	224–264	0.734	
R: GTTGTTGTTCTTCCCTGCCC		
Cmi31	F: ACGTCACTCACTACACCCCG	(ATT)n	218–260	0.708	
R: AGCTAGCTCAGCCTGGATGC		
Cmi32	F: CAGGGTGTATCCTGCCTTCG	(ATT)n	239–269	0.782	
R: GCCGTCCATGAAATCTGGC		
Cmi34	F: CCAGAAATTGAGCCTGACCC	(TCGGG)n	275–350	0.549	
R: AACCTGGGGTTTAAATCGGG		
Cmi35	F: TCAGCAAATCATACGGCTGG	(ATT)n	296–338	0.769	
R: CTGGCCATTCTCTGTTGGC		
Cmi36	F: CAGACATCATTGTTGGCCCC	(ATT)n	148–175	0.843	
R: GTCCAAGCACGAATCCAGC		
Cmi38	F: CATCATGCATAAAAGTGCCCC	(ATCT)n	180–260	0.912	
R: TTGGAGCTTAGATGCCTTGC		
Cmi39	F: GTCTGTTCATGCGCACTTCC	(TTC)n	160–244	0.932	
R: AATTAGTGCTCAGGGGTGGG		
Cmi40	F: TGTTGAGCATGAATGAGTCGG	(AGTG)n	189–313	0.946	
R: CCTATGAGCCCTATGAACACTGG		
Cmi41	F: CGGACTATTAAAGTACAGTGTGAAAGG	(ATCT)n	254–334	0.886	
R: CAGTGATACCCAGTCCACCG		
Cmi42	F: GAAGGTAATCTGGGTCAGCAGG	(AAC)n	273–306	0.706	
R: CTTGGTGGAAGACCCTCAGC		
Cmi44	F: GGTTCCTAAATGGTTCTTAGATTTGC	(ATAGT)n	235–300	0.706	
R: TGAGCATCCTGCACATTTCC		
Cmi45	F: TGATTTGCCAGTGTAATGAGAGG	(ATT)n	278–305	0.800	
R: CATGGTTTTAATCTTATAATCAGCCG		
Cmi46	F: AAATGGAGCCAACTTAGCCG	(ATT)n	148–169	0.658	
R: GCAACTGTATCACCCAAACTACC		
Cmi47	F: GGAGTCTAACGGAGGGGAGC	(ATT)n	149–194	0.827	
R: CAAGTACCTAATAAAATGGCCGC		
Cmi48	F: GGATGGGTACAAACGAAGGG	(ATT)n	226–256	0.778	
R: TGTGCAGGTGTGGTTTACCG		
Cmi49	F: GTTGCTCTGTACCACTCACCG	(ATT)n	105–132	0.679	

Table 2 Genetic diversity per locus and across 20 loci in C. mivartii from Colombian rivers.

Site (N)		Cmi31	Cmi40	Cmi46	Cmi44	Cmi41	Cmi39	Cmi03	Cmi47	Cmi32	Cmi36	Cmi38	Cmi49	Cmi45	Cmi09	Cmi01	Cmi17	Cmi08	Cmi16	Cmi11	Cmi12	Across loci	
Man(24)	Na	11.000	17.000	6.000	9.000	14.000	15.000	6.000	9.000	6.000	10.000	16.000	7.000	9.000	7.000	16.000	6.000	8.000	10.000	12.000	9.000	10.150	
HO	0.750	0.792	0.583	0.667	0.917	0.958	0.750	0.917	0.667	0.917	0.625	0.833	0.875	0.792	0.917	0.667	0.667	0.875	0.583	0.708	0.773	
HE	0.785	0.931	0.699	0.729	0.912	0.926	0.767	0.871	0.778	0.878	0.930	0.771	0.849	0.733	0.936	0.762	0.736	0.813	0.887	0.850	0.810	
P	0.657	0.001	0.558	0.419	0.496	0.873	0.608	0.175	0.155	0.722	0.000	0.719	0.671	0.791	0.727	0.425	0.470	0.396	0.000	0.175	0.000	
F	0.025	0.131	0.147	0.066	−0.026	−0.056	0.001	−0.074	0.125	−0.067	0.314	−0.103	−0.052	−0.103	0.000	0.106	0.075	−0.099	0.328	0.149	0.044	
Grande(30)	Na	8.000	23.000	7.000	9.000	10.000	17.000	6.000	8.000	7.000	10.000	16.000	8.000	10.000	7.000	18.000	6.000	7.000	10.000	12.000	9.000	10.400	
HO	0.733	0.867	0.733	0.733	0.700	0.933	0.567	0.800	0.800	0.767	0.833	0.467	0.733	0.767	0.900	0.767	0.700	0.833	0.667	0.733	0.752	
HE	0.750	0.959	0.751	0.685	0.888	0.946	0.720	0.839	0.768	0.872	0.931	0.736	0.840	0.735	0.914	0.756	0.745	0.859	0.877	0.799	0.805	
P	0.739	0.001	0.479	0.553	0.290	0.798	0.050	0.744	0.744	0.229	0.085	0.001	0.244	0.354	0.255	0.557	0.867	0.316	0.025	0.479	0.000	
F	0.005	0.081	0.008	−0.088	0.198	−0.003	0.199	0.030	−0.059	0.106	0.090	0.355	0.112	−0.061	−0.002	−0.031	0.045	0.013	0.227	0.067	0.065	
Las Culebras(29)	Na	8.000	21.000	6.000	10.000	13.000	23.000	6.000	10.000	8.000	9.000	14.000	8.000	10.000	5.000	19.000	8.000	9.000	10.000	12.000	9.000	10.900	
HO	0.759	0.897	0.655	0.621	0.724	0.897	0.759	0.862	0.793	0.793	0.793	0.759	0.724	0.586	0.793	0.793	0.793	0.759	0.759	0.621	0.757	
HE	0.716	0.947	0.718	0.690	0.915	0.943	0.663	0.862	0.852	0.812	0.918	0.717	0.798	0.673	0.944	0.772	0.786	0.793	0.843	0.783	0.793	
P	0.667	0.021	0.200	0.122	0.166	0.066	0.839	0.820	0.713	0.091	0.055	0.582	0.074	0.166	0.024	0.006	0.613	0.277	0.029	0.372	0.000	
F	−0.078	0.037	0.072	0.085	0.195	0.032	−0.164	−0.018	0.053	0.006	0.121	−0.077	0.077	0.113	0.145	−0.045	−0.027	0.026	0.084	0.194	0.041	
La Raya(19)	Na	10.000	18.000	6.000	7.000	10.000	18.000	5.000	9.000	6.000	8.000	14.000	7.000	9.000	5.000	16.000	8.000	6.000	10.000	10.000	11.000	9.650	
HO	0.947	0.895	0.632	0.526	0.632	0.947	0.526	0.789	0.789	0.789	0.842	0.632	0.737	0.842	0.842	0.526	0.579	0.737	0.737	0.737	0.734	
HE	0.835	0.952	0.740	0.654	0.859	0.950	0.764	0.849	0.792	0.849	0.929	0.741	0.846	0.741	0.942	0.718	0.733	0.842	0.885	0.871	0.803	
P	0.863	0.514	0.391	0.189	0.022	0.529	0.050	0.313	0.832	0.890	0.133	0.121	0.563	0.043	0.142	0.042	0.063	0.417	0.161	0.303	0.000	
F	−0.165	0.034	0.123	0.174	0.245	−0.024	0.292	0.045	−0.023	0.045	0.069	0.125	0.106	−0.167	0.082	0.248	0.188	0.101	0.145	0.131	0.089	
Panela(35)	Na	9.000	21.000	7.000	9.000	10.000	18.000	4.000	11.000	8.000	9.000	17.000	9.000	9.000	6.000	21.000	7.000	8.000	9.000	14.000	10.000	10.800	
HO	0.771	0.914	0.543	0.743	0.657	0.886	0.657	0.857	0.886	0.886	0.800	0.571	0.800	0.657	0.829	0.714	0.800	0.857	0.771	0.714	0.766	
HE	0.732	0.952	0.691	0.788	0.893	0.932	0.729	0.842	0.785	0.864	0.917	0.697	0.813	0.645	0.918	0.771	0.764	0.835	0.887	0.814	0.802	
P	0.416	0.843	0.073	0.718	0.001	0.233	0.446	0.877	0.721	0.586	0.012	0.012	0.605	0.240	0.020	0.292	0.923	0.297	0.256	0.076	0.000	
F	−0.069	0.025	0.203	0.044	0.253	0.036	0.085	−0.033	−0.145	−0.040	0.115	0.168	0.002	−0.033	0.085	0.061	−0.063	−0.042	0.118	0.109	0.044	
Chucurí(35)	Na	8.000	21.000	7.000	10.000	16.000	17.000	6.000	10.000	7.000	9.000	15.000	9.000	9.000	6.000	19.000	6.000	7.000	10.000	14.000	12.000	10.900	
HO	0.600	0.771	0.486	0.771	0.857	0.886	0.714	0.771	0.800	0.686	0.800	0.714	0.686	0.629	0.886	0.743	0.771	0.800	0.857	0.629	0.743	
HE	0.680	0.953	0.688	0.733	0.904	0.926	0.704	0.838	0.798	0.881	0.920	0.693	0.817	0.705	0.944	0.787	0.775	0.807	0.892	0.809	0.801	
P	0.293	0.000	0.025	0.750	0.192	0.580	0.323	0.069	0.907	0.005	0.019	0.544	0.000	0.435	0.269	0.233	0.477	0.496	0.891	0.003	0.000	
F	0.104	0.179	0.284	−0.068	0.038	0.030	−0.029	0.066	−0.018	0.211	0.118	−0.045	0.148	0.096	0.048	0.042	−0.010	−0.006	0.025	0.212	0.071	
Puerto Berrío(35)	Na	11.000	21.000	6.000	11.000	14.000	19.000	6.000	10.000	9.000	9.000	14.000	7.000	8.000	6.000	16.000	6.000	8.000	11.000	13.000	8.000	10.650	
HO	0.514	0.971	0.657	0.743	0.886	0.800	0.829	0.857	0.829	0.971	0.657	0.714	0.800	0.686	0.914	0.800	0.714	0.771	0.914	0.400	0.771	
HE	0.715	0.952	0.688	0.746	0.907	0.923	0.722	0.863	0.842	0.850	0.921	0.617	0.819	0.748	0.931	0.782	0.740	0.860	0.875	0.628	0.795	
P	0.003	0.652	0.781	0.476	0.604	0.008	0.226	0.172	0.707	0.542	0.003	0.825	0.314	0.499	0.536	0.978	0.679	0.254	0.344	0.001	0.002	
F	0.270	−0.036	0.031	−0.011	0.010	0.121	−0.164	−0.008	0.001	−0.159	0.276	−0.175	0.010	0.070	0.004	−0.038	0.021	0.090	−0.060	0.354	0.030	
Notes:

Ra, allelic size range; Na, average number of alleles per locus; HO and HE, observed and expected heterozygosity, respectively; P, statistical significance for tests of–departure of Hardy–Weinberg equilibrium after Bonferroni correction.

Values in bold denote significant departures from Hardy-Weinberg equilibrium.

The pairwise tests of genotypic disequilibrium were nonsignificant and no evidence of null alleles or scoring errors were detected by Micro-Checker in the overall sample. However, five of the 27 loci (Cmi02, Cmi05, Cmi07, Cmi34 and Cmi48) showed the lowest PIC values (Table 1), departure of allelic frequencies from Hardy–Weinberg equilibrium expectations in three or more of the evaluated samples and inconsistencies in the amplification (Table 2); consequently, they were excluded from additional analysis. Furthermore, the BayeScan analyses showed significant evidence of putative signals of diversyifing/positive selection for the additional loci Cmi35 (posterior probability: 0.946; log10PO: 1.242; alpha: 1.339 and PSimul Fst < sample Fst: 0.032) and Cmi42 (posterior probability: 0.999; log10PO: 3.699; alpha: 1.670 and PSimul Fst < sample Fst: 0.042) in the comparison between the floodplain lake Las Culebras and the remaining localities.

Results from the Bottleneck tests (Table 3) were significant for all populations under the infinite alleles model (IAM) and some populations under the two-phase model (TPM); whereas they were non-significant under the stepwise mutation model (SMM). Due to the thought that few loci follow the strict SMM (Piry, Luikart & Cornuet, 1999), the best estimation of HE at mutation-drift equilibrium is expected under a combination of IAM and TPM. Additionally, all values of the M ratio were substancially smaller than 0.68, indicating that all populations have experienced recent and severe reduction in population size (Table 3).

Table 3 Bottleneck test for C. mivartii from some sectors of the Colombian rivers Cauca and Magdalena.

Site	IAM	SMM	TPM	M ratio	
Man	0.000	0.663	0.038	0.207 ± 0.078	
Grande	0.000	0.869	0.005	0.206 ± 0.082	
Las Culebras	0.000	0.997	0.324	0.226 ± 0.074	
La Raya	0.000	0.727	0.022	0.194 ± 0.075	
Panela	0.000	0.980	0.041	0.213 ± 0.069	
Chucurí	0.000	0.934	0.101	0.218 ± 0.079	
Puerto Berrío	0.000	0.774	0.131	0.214 ± 0.072	
Notes:

Expected heterozygosity excess is presented as P-values from the Wilcoxon sign-rank test using the infinite alleles model (IAM), stepwise mutation model (SMM) and two-phase model (TPM). M ratio, mean ratio of the number of alleles compared with the range in allele size.

Values in bold denote statistical significance.

The Bayesian analysis, discriminant analysis of principal components (Figs. 3A and 3B), AMOVA (F′ST = 0.001; P = 0.285) and pairwise comparisons with standardized estimators F′ST and Jost’s Dest (Table 4), evidenced the presence of a single genetic stock. Consequently, the Mantel test showed a weak spatial correlation of genetic distances in the evaluated sector (R2 = 0.001; P = 0.031).

Figure 3 Population structure suggested by STRUCTURE (A) and discriminant analysis of principal components (B).

Bar plot of population ancestry coefficients estimated by STRUCTURE is provided for K = 2 and Q-matrixes were consensus estimates produced by CLUMPP across 20 iterations of STRUCTURE.

Table 4 Pairwise Jost’s Dest (upper diagonal) and F′St (below diagonal) among samples of C. mivartii from the Colombian Magdalena-Cauca basin.

	Man	Grande	Las Culebras	La Raya	Panela	Chucurí	Puerto Berrío	
Man		−0.005	0.022	−0.001	0.008	−0.007	0.010	
Grande	0.010		0.016	−0.007	−0.011	−0.024	−0.007	
Las Culebras	0.013	0.011		0.013	0.020	0.007	0.016	
La Raya	0.013	0.011	0.013		0.001	−0.005	0.021	
Panela	0.010	0.007	0.011	0.011		−0.010	0.006	
Chucurí	0.009	0.006	0.009	0.011	0.007		−0.005	
Puerto Berrío	0.010	0.007	0.010	0.013	0.008	0.007		
Note:

Values were not statistically significant after the Bonferroni correction.

Discussion

This study tested the hypothesis that C. mivartii exhibits genetically structured populations, according to an isolation by distance model; to accomplish that, a set of primers was developed for the amplification of 27 loci microsatellites, 22 of which exhibited allelic frequencies according to Hardy–Weinberg equilibrium expectations in most of the evaluated samples. This work also showed evidence of putative selection in 9.091% of the 22 loci examined, which is in line with the percentage of outlier loci (5–18%) reported for microsatellites in migratory marine fishes (Larsson et al., 2007; Rhode et al., 2013; Liu et al., 2016) and different molecular markers in other taxa (for reviews see Nosil, Funk & Ortiz-Barrientos (2009)).

The outlier loci found in this study may represent false positives resulting from the inclusion of severely bottlenecked populations (Teshima, Coop & Przeworski, 2006; Foll & Gaggiotti, 2008), although the significant excess of heterozygosity and small M ratio values were found even in populations that did not exhibit outlier loci. Alternatively, the outlier loci may result from asymmetry gene flow by unidirectional migration (Hansen, Meier & Mensberg, 2010) as well as hitchhiking selection resulting from temporal disconnections between the floodplain lake Las Culebras and the Caribona river main stream. Although the contribution of these events in the observed findings remains unaddressed, both explanations are plausible considering the positive/negative ENSO successions observed in the last decade for the Magdalena-Cauca basin, the strongest recorded so far for this basin in terms of low water levels and high temperatures (IDEAM—Instituto de Hidrología Meteorología y Estudios Ambientales, 2014).

As this is the first report of population genetics for a species of the Curimatidae family, the results of this study were compared with species of the phylogenetically related family, Prochilodontidae. In C. mivartii values of the average number of alleles per locus (10.493) and expected and observed heterozygosities (HO = 0.757 and HE = 0.801) were similar to those found in studies that used species-specific microsatellite loci with large repeat motifs such as Prochilodus argenteus (Sanches et al., 2012; Melo et al., 2013) and Ichthyoelephas longirostris (Landínez-García & Márquez, 2016).

In contrast to the a priori expectation of a population structure concordant with an isolation by distance model, this study evidenced a high genetic connectivity of C. mivartii even in localities separated by over 350 km (Man river and Puerto Berrío). Given that this species is considered a short-distance migrant (<100 km; Zapata & Usma, 2013), a decrease in genetic similarity is expected among populations as the geographic distance between them increases. However, allele frequencies found in this study were not spatially autocorrelated at distances two and three times longer than the estimated migration range of C. mivartii, providing no support to our hypothesis.

The above findings may indicate that the dispersion range of the species is underestimated as the only register (10.1 km) published contained information about the recapture of a single individual out of 149 marked, using the mark-recapture method (López-Casas et al., 2016). Consequently, this outcome suggests that C. mivartii exhibits at least a medium migration range, a category that includes a displacement capacity of between 100 and 500 km (Zapata & Usma, 2013).

An alternative and non-excluding explanation is that C. mivartii displacements are performed in various events and in different directions and magnitudes, thereby providing a possible explanation for the high levels of connection among floodplain lakes along the studied range. This may also imply that C. mivartii does not exhibit a homing behavior like that described in various members of the Prochilodontidae family (Godinho & Kynard, 2006).

Similarly, given the sample collection period, the extensive gene flow might have resulted from a strong 2010/2011 La Niña event, which affected precipitation patterns world-wide (Boening et al., 2012) and caused damage in Colombia associated with occurrences of floods, windstorms, lightning and landslides (Hoyos et al., 2013). This event, preceded by warm anomalies during the first quarter of 2010 (Hoyos et al., 2013), may also explain an apparent excess of novel alleles and an incomplete allele frequency distribution, as evidenced by the Bottleneck analysis.

In summary, rejecting the hypothesis of isolation by distance, C. mivartii represents a single stock in the lower section of the Colombian Magdalena-Cauca basin, which exhibits high genetic diversity and an apparent recent and severe reduction in population size. This information constitutes a baseline for monitoring the population genetics of these species that inhabit main streams of the rivers and floodplain lakes downstream from the Ituango hydropower construction. Additionally, this knowledge is crucial to establish conservation units and facilitate its management.

Conclusions

The results of this study indicate that populations of C. mivartii are not structured according to an isolation by distance model in a sector three times longer than its estimated migration range (<100 km). This study also developed a group of 20 loci microsatellites, the first report for the Curimatidae family, that can be used in subsequent studies to delve into the conservation, basic genetics and biology of the species. Additionally, we found two outlier loci putatively under selection that will require further assessment in estimating their usefulness in future studies. These markers may be valuable for delineating conservation units known as Management, Adaptive and Evolutionarily Significant Units (Funk et al., 2012), thereby providing a more complete insight into the evolutionary potential for conservation of wild populations and the short- and long-term persistence of the species in the Magdalena-Cauca basin, a historically neglected aspect among most Neotropical fish species.

Supplemental Information

Supplemental Information 1 Raw sequences reads of 27 microsatellite loci selected for Curimata mivartii.

Click here for additional data file.

Supplemental Information 2 Raw sequences reads of 27 microsatellite loci selected for Curimata mivartii .

Click here for additional data file.

Supplemental Information 3 Genotype data at 27 microsatellite loci developed for the freshwater fish Curimata mivartii in GenAlex format.

First line indicates respectively: Number of loci, Number of individuals, Number of sampled site, Number of sample size for each sampled site, number of regions, number of individuals by region. Second line indicates respectively: sample identification (ID), Site, Locus name for 27 loci. Loci highlighted in gray were excluded of further population genetic analysis.

Click here for additional data file.

Supplemental Information 4 Genotype data at 20 microsatellite loci included in the population genetic analysis of the freshwater fish Curimata mivartii in GenAlex format.

First line indicates respectively: Number of loci, Number of individuals, Number of sampled site, Number of sample size for each sampled site, number of regions, number of individuals by region. Second line indicates respectively: sample identification (ID), Site, Locus name for 27 loci.

Click here for additional data file.

Supplemental Information 5 Electropherograms of 27 microsatellite loci for the freshwater fish Curimata mivartii.

Click here for additional data file.

The authors would like to thank the Centro Nacional de Enunciation Genómica, Universidad de Antioquia (Medellín, Colombia), for their assistance with bioinformatics analysis and José Gregorio Martínez for his collaboration in the analysis of the putative signal of diversifying/positive selection loci. The authors also thank to Cesar Amaral and the anonymous reviewers for their comments that improved the final version of this article.

Additional Information and Declarations

Competing Interests

Author Contributions

Ethics

Data Availability

The authors declare that they have no competing interests.

Ricardo M. Landínez-García conceived and designed the experiments, performed the experiments, analyzed the data, contributed reagents/materials/analysis tools, prepared figures and/or tables, authored or reviewed drafts of the paper, approved the final draft.

Edna J. Marquez conceived and designed the experiments, analyzed the data, contributed reagents/materials/analysis tools, prepared figures and/or tables, authored or reviewed drafts of the paper, approved the final draft.

The following information was supplied relating to ethical approvals (i.e., approving body and any reference numbers):

The preserved tissues of this fishery resource were provided by Integral S.A. (Scientific cooperation agreement between Universidad Nacional de Colombia and Integral S.A., on 19th September 2013).

The following information was supplied regarding data availability:

The raw data is included in the Supplemental Files.

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
