# Peer review of "Microsatellite loci development and population genetics in Neotropical fish Curimata mivartii (Characiformes: Curimatidae)"

_PeerJ, doi:10.7717/peerj.5959_

## Round 0.1 · original submission · Major Revisions

In accordance with reviewer#1, I think that raw data accessibility is crucial to evaluate the quality of the genotyping, in particular in a manuscript claiming “microsatellite loci development”. By the way, it would more correct to state ‘development of microsatellite markers’.
I further question the following points:

- I could not find the Supplementary material cited at lines 147-8, 213.
- The table Curimata_mivartii_Microsatellite_genotypes.xlsx (Supplemental Files) must be carefully revised (e.g. ‘144’ in the heading row) and a legend provided. Besides there are too many missing genotypes (0s) and (apparent) homozygotes, which may indicate amplification failures (exceedingly frequent silent alleles) in accordance with the frequent HWE significant departures observed. This lack of robustness on genotyping compromises all the conclusions, namely on the population structure.
- Legends for tables and figures are missing or incomplete.

·

Basic reporting

Yes, the MS is clear and presents sufficient field background. It is also well structured but it presents some issues related with the raw data availability and figures.

The raw data is only partially available. They should be made available for scrutiny since they complement/validate their analyses.

The figures are relevant for the discussion but some issues should be addressed:

- In Figure 1, the small map of Colombia presented on Fig, 1B should be changed to Fig.1A for clarity and better visualization. They should also change the figure caption.

- The manuscript lacks an image of the species. Since it deals with one unique species, it is important for the reader to visualize the species the MS is talking about.

- The manuscript does not present any raw data from the loci analyzed. The authors should present the raw data (electropherogram) since the MS also deals with the development of the microsatellites.

Experimental design

The MS is within the scope of the journal and the research question is clear. The methods are correctly used and the research is relevant since it deals with the Curimatidae, a clade without any kind of paper on this matter.

I failed to find an ethics statement and cannot precise if the research was conducted following high quality standards. The unique statement provided is related with the agreement between Universidad Nacional de Colombia and Integral S.A., on 19th September 2013)… nothing presented about fishing permissions and sample gathering methods.

Validity of the findings

The main goal here is the development of microsatellites for the Curimatidae, a clade without any kind of previous studies on this field. The authors made it although some issues should be addressed on the discussion and conclusions.

The MS hypothesized about isolation by distance but the authors poorly discussed this hypothesis in the light of their results… they also didn’t presented nothing about this in the final conclusions. If isolation by distance is the hypothesis for the MS, they should be addressed within the MS conclusions.

A second issue is related with the discussion about the ENSO. Imho, the authors cannot use ENSO to explain what they didn’t observed from their results. It seems a bit speculative for me since the genetic differentiation was not significant after the exclusion of both loci putatively under selection for Las Culebras. If they don’t have differentiation, why use ENSO to explain the genetic structure that was not observed?

Additional comments

Some minor comments here:

The authors should check the journal standard for using “et al.” and verify the MS correcting it along the entire MS. Sometimes “et al.” is used sometimes the authors use the entire author list of the manuscript. For instance, check lines 38, 41, and 49 (and also the rest of the MS).

Line 36 - Please add a reference after “… South America.”

Line 68 – Remove the period marker after Barandica

Line 71 – I suggest the use of standard scientific nomenclature when referring the species rather than “vizcaína”

Line 138 - Check number format 5000 / 200,000 / 50,000

Line 213 – I’m not sure about the validity of the comparison with microsatellites from migratory marine fishes. Consider to remove this lines.

Line 224 – a priori, check Italics

Line 272 – Evolutionary potential? Please consider a review here.

Line 273 – “… in the main …”. It seems lost here. Please verify.

Reviewer 2 ·

Basic reporting

I identified lack of attention in writing the manuscript. I made some observations and suggestions below:
1. Did you use next-generation sequencing? Although it is listed as a key word, no results were shown.

2. The language could be improved, particularly in the introduction and in the discussion. Please check carefully the text again.

3. The introduction is quite informative and appropriate, but the references cited in the text are not in the PeerJ norms:
a) For example, lines 38 and 52 (For four or more authors, abbreviate with ‘first author’ et al. and No comma before &)
b) Multiple references to the same item should be separated with a semicolon (;) and ordered chronologically
c) References: are not in accordance with the norms. Check carefully the references.

4. Senseless words in the text “de novo” lines 72-73. De novo is a method of sequencing, but I think it was not used in this study.

Results:
5. How many clones were sequenced? There isn't this information in the ms

6. GenBank accession numbers: There isn't this information in the ms


7. Figure 1: Figure with the delimitation of the basin should be transferred to 1A from 1B. The designation of A and B should appear in the legend.

8. Figure 3: The figures 3b and 3d should be removed (If you are sure that loci outliers are not a technical artifact, you must discuss these results further. The identification of loci under divergent selection is a fundamental step in understanding the evolutionary process because these loci are responsible for the genetic variations that affect fitness in different environments. Understanding how environmental forces give rise to adaptive genetic variation is a major challenge).

Experimental design

1. The ms fits the standards required for original research articles.
2. The ms is a good example of a well conducted study focused on answering an objective question that is important to conservation of fish fauna in the studied region.
3. The present ms brings the first report of population genetics for a species of the Curimatidae family. These information can be used in the conservation of this species, providing a more complete insight into the evolutionary potential and the short-/long-term persistence in the Magdalena-Cauca basin.

4. Methods was not described in sufficient detail and information to allow replication. Some information was missing, such as:
a) The DNA extraction protocol is according to what author?
b) How many ng of DNA were required for the construction of the genomic library?

5. Was the sampling protocol approved by the Ethics Committee on the Use of Animals? There isn't this information in the ms.

Validity of the findings

No comment

Additional comments

The ms is a good example of a well conducted study focused on answering an objective question that is important for conservation of fish fauna in the studied region. However, I identified lack of attention in writing the manuscript. Thus, I recommend checking carefully some parts of the text and reorganization of figures.

Reviewer 3 ·

Basic reporting

The manuscript by Landínez-García and Marquez describes a new set of microsatellite loci in Neotropical fish Curimata mivartii. The article is well written, with sufficient references. The approach is well explained and a good example for other authors to follow.

Experimental design

The aim of the work is important. The analyses are performed with a high technical standard and described in sufficient detail. The methods are well described. A few comments:
The authors should mention how the DNA was extracted.
PCR reactions is redundant
Why the need of using so many (42) PCR cycles in fresh samples?
The authors should have sequenced the amplicons in a few samples to confirm the STR structure.

Validity of the findings

The data is robust and statistically sound. I suggest a PC or MDS analyses to visualize the estimated genetic distances. The conclusion are well stated.

---

## Round 0.2 · Minor Revisions

All questions were answered and recommendations were accepted by the authors.

The article is almost accepted, however as noted by the Section Editors you do not currently provide your raw (neither the allelic data nor the sequences from which the microsat loci were discovered and from which primers were developed). This does not meet our data & materials sharing policy (https://peerj.com/about/policies-and-procedures/#data-materials-sharing ), and should be corrected prior to publication. You can either use some federated database like GenBank or DataDryad, or you can include the data as a supplementary file, but our policies do require raw data sharing before your article can be Accepted.

·

Basic reporting

Almost all previous issues were addressed/answered by the authors and as far as I am concerned, the manuscript is good to go.

Experimental design

Please check the section 1. Basic Reporting

Validity of the findings

Please check the section 1. Basic Reporting

Reviewer 2 ·

Basic reporting

After the MS review, all questions were answered and recommendations were accepted by the authors. However, microsatellites loci were not deposited in GenBank. That way, I suggest that the authors make the deposit.
The MS fits the standards required for original research articles.

Experimental design

The analyses are performed with a high technical standard and described in sufficient detail.

Validity of the findings

The present MS brings the first report of population genetics for a species of the Curimatidae family. These information can be used in the conservation of this species, providing a more complete insight into the evolutionary potential and the short-/long-term persistence in the Magdalena River basin.

Additional comments

The MS is a good example of a well conducted study focused on answering an objective question that is important to conservation of fish fauna in the studied region.

Reviewer 3 ·

Basic reporting

n/a

Experimental design

n/a

Validity of the findings

n/a

Additional comments

n/a

---

## Round 0.3 · accepted · Accept

Recommendations were accepted by the authors and raw data provided.

#